# Therapeutic Potentials of Extracellular Vesicles for the Treatment of Diabetes and Diabetic Complications

**DOI:** 10.3390/ijms21145163

**Published:** 2020-07-21

**Authors:** Wei Hu, Xiang Song, Haibo Yu, Jingyu Sun, Yong Zhao

**Affiliations:** 1Center for Discovery and Innovation, Hackensack Meridian Health, Nutley, NJ 07110, USA; whu2@stevens.edu (W.H.); Xiang.Song@HMH-CDI.org (X.S.); Haibo.Yu@HMH-CDI.org (H.Y.); 2Department of Chemistry and Chemistry Biology, Stevens Institute of Technology, Hoboken, NJ 07030, USA; Jsun20@stevens.edu

**Keywords:** extracellular vesicle, exosomes, isolation, cell-to-cell communication, biomarker, diabetes, diabetic complication

## Abstract

Extracellular vesicles (EVs), including exosomes and microvesicles, are nano-to-micrometer vesicles released from nearly all cellular types. EVs comprise a mixture of bioactive molecules (e.g., mRNAs, miRNAs, lipids, and proteins) that can be transported to the targeted cells/tissues via the blood or lymph circulation. Recently, EVs have received increased attention, owing to their emerging roles in cell-to-cell communication, or as biomarkers with the therapeutic potential to replace cell-based therapy. Diabetes comprises a group of metabolic disorders characterized by hyperglycemia that cause the development of life-threatening complications. The impacts of conventional clinical treatment are generally limited and are followed by many side effects, including hypoglycemia, obesity, and damage to the liver and kidney. Recently, several studies have shown that EVs released by stem cells and immune cells can regulate gene expression in the recipient cells, thus providing a strategy to treat diabetes and its complications. In this review, we summarize the results from currently available studies, demonstrating the therapeutic potentials of EVs in diabetes and diabetic complications. Additionally, we highlight recommendations for future research.

## 1. Introduction

Diabetes is a major public health issue with complex etiology affecting over 350 million people worldwide. By 2045, its incidence is estimated to increase to 700 million [1]. Diabetes is the sixth leading cause of death in the US and is associated with increased risks for heart disease, stroke, kidney disease, blindness, and amputations [2,3,4]. Classifications are divided into three categories, namely type 1 diabetes (T1D), type 2 diabetes (T2D), and gestational diabetes [5]. The most common diagnosis is T2D, accounting for 90% of diabetic subjects worldwide [6]. T2D is characterized mainly by insulin resistance, aberrant production of insulin, and chronic low-grade inflammation in peripheral tissues, including adipose tissue, the liver, and muscle [7]. T1D is caused by a shortage of insulin-producing cells due to the autoimmune destruction of pancreatic islet β cells [8]. Gestational diabetes is a common metabolic disease that occurs during pregnancy, with variable degrees of glucose intolerance [9]. Current therapeutic methods to treat diabetes typically include oral hypoglycemic drugs and insulin; however, they are not a real “cure” for diabetes and not effective for improving a patient’s condition. Therefore, the need for alternative therapies for diabetes patients is urgent.

Cell-based therapy is an alternative method of diabetes treatment. Researchers have used stem cells or immune cells to treat diabetes. Extracellular vehicles (EVs) are defined as phospholipid-bilayer-enclosed vesicles carrying bioactive receptors, lipids, proteins, and nucleic acids that interact with target cells, driving the subsequent modification of target cells. EVs are released by most cell types and recently have demonstrated to not only act as promising biomarkers for disease but also as therapeutic agents for certain diseases [10,11]. Consequently, EVs secreted by stem cells or immune cells are receiving increased research attention. Researchers have exhibited that EVs have strong therapeutic potential by delivering their cargo into target cells and acting on different signaling pathways [12,13,14]. EVs have become established as signaling mediators between cells, including islet cells, though they have only recently have gained popularity as a candidate for diabetes treatment. This review discusses current advances in the utilization of EVs as a treatment for diabetes and diabetic complications.

## 2. Extracellular Vesicles

EVs are one of the fast-growing areas in biomedical research and clinical translational medicine. For the detailed information on EVs’ cell biology, biogenesis, secretion, and intercellular interactions, readers are encouraged to visit excellent review articles [10,15,16]. The immune modulation and potential clinical applications of mesenchymal stem cell (MSC)-derived EVs and others have been reviewed elsewhere [17,18,19,20,21]. This review provides a brief introduction to EVs.

### 2.1. Classification and Origin

Extracellular vesicles are a heterogeneous population of small membrane vesicles (30–2000 nm) released from different types of activated or apoptotic cells. Based on their size and origin (Table 1), EVs have been classified into three major groups: exosomes, microvesicles (MVs), and apoptotic bodies [22,23] (Figure 1 and Table 1).

Exosomes are derived from the endocytic compartment and range from 30 to 100 nm in size. Specifically, the cell’s plasma membrane is internalized to generate an early endosome. Next, intraluminal vesicles (ILVs) pinch the endosomal limiting membrane inward and bud into the endosome. Selected proteins and RNAs are then packed into the ILVs by the endosomal sorting complex required for transport (ESCRT)-dependent machinery or ESCRT-independent machinery. The endosome is now forming a multivesicular body (MVB). Consequently, a partial number of MVBs are digested by fusion with lysosomes, while others fuse with the plasma membrane through unknown mechanisms involving RAB-27 and soluble NSF (N-ethylmaleimide-sensitive factor) attachment protein receptor (SNARE) proteins. These payload ILVs are secreted into the extracellular medium as exosomes [16]. Although the specific mechanism is still not well-known, it appears that growth factor will promote the development of MVBs. The cell modifies its production of exosome along with its needs [31]. Exosomes are surrounded by a bilayer of phospholipids enriched with ceramide and cholesterol. The surface molecules anchored in the exosome’s membrane include adhesion molecules (integrins and intracellular adhesion molecules (ICAMs)), tetraspanin proteins (CD9, CD63, and CD81), and immunity associated molecules (major histocompatibility complex (MHC)-I and MHC-II). The cytoplasm of an exosome includes ESCRT-related proteins (apoptosis-linked gene 2-interacting protein X (ALIX) and tumor susceptibility 101 (TSG101)), RNAs (mRNA), microRNAs (miRNAs), long noncoding RNAs (lncRNAs), cytoskeletal proteins (actin and tubulin) and metabolic enzymes (glyceraldehyde-3-phosphate dehydrogenase (GAPDH) and ATPase) [25,32,33]. Furthermore, exosomes contain some specific molecules that depend on their original cells. For example, researchers discovered exosomal MHC molecules, released by dendritic cells, interacting with T cells, to induce antitumor immunity [34].

### 2.2. Isolation and Characterization

Although MVs and exosomes are produced by different mechanisms, most isolation methods do not isolate a pure population because of their size overlap. So far, several EV isolation methods have been established, including ultracentrifugation, density gradient centrifugation, ultrafiltration, immunoprecipitation, polymer-based precipitation, size exclusion chromatography (SEC), and microfluidic device isolation [35,36,37,38]. (1) Ultracentrifugation isolation is based on the size of the EVs, typically consisting of a sequential increase in centrifugal force to pellet cells and debris (<2000× *g*), large EVs (10,000–20,000× *g*), and small EVs (100,000–200,000× *g*). (2) Density gradient centrifugation isolation depends upon the size and mass density to isolate EVs. (3) Ultrafiltration isolation is based on the size of the EVs, in which samples are passed through a membrane with specific pore size by pressure or centrifugation. (4) Polymer-based precipitation is based on the application of a polymer solution, such as polyethylene glycol (PEG), to minimize EVs’ solubility and drive their precipitation. (5) Immunoprecipitation isolation applies monoclonal antibodies immobilized on the surface of a plate or beads to capture the EVs. (6) Size exclusion chromatography depends on the size of the EVs, which are separated by using a column. The majority of the EVs is eluted before the soluble components. (7) Microfluidic device isolation depends on the designed device isolating a subpopulation of vesicles based on their properties such as size, density, or specific markers. However, these existing isolation methods remain challenging, as each one has its advantages and disadvantages (Table 2).

After isolation, EV populations need to be characterized for their intended downstream application, for which a variety of technologies are available. Size and protein markers are key characteristics of EVs. Using transmission electron microscopy (TEM), scanning electron microscopy (SEM), nanoparticle tracking analysis (NTA), and dynamic light-scattering (DLS), we can determine the size of EVs [41]. Western blotting and enzyme-linked immunosorbent assays are convenient methods to characterize the EVs, using some of the EV classical biomarkers such as CD9, CD63, CD81, ALIX, and TSG101 [42,43]. These methods are applied to identify and characterize EVs in combination. Currently, flow cytometry has emerged as an attractive method for EVs analysis. Most of the current flow cytometry technologies can detect particles larger than 500 nm. Therefore, the EVs must be bound to antibodies or sulfate latex beads with a size that is in the detectable size ranges of the flow cytometer [44]. The latest flow cytometry technology (termed nanoscale flow cytometry) is highly sensitive and allows for the direct analysis of single EV and protein profile down to 40 nm [45]. Furthermore, if we use the identified markers on a single EV, nanoscale flow cytometry will enable the characterization of novel specific subpopulations of EVs and the diagnostic markers on EVs. This characterization might advance the understanding of EV biology and enable early diagnosis of diseases [46]. Therefore, nanoscale flow cytometry will become a powerful technique for future EV research and disease diagnosis.

### 2.3. EV Interactions with Target Cells

The function or therapeutic potential of EVs depends on their ability to interact with target cells. There are various mechanisms by which EVs interact with target cells (Figure 2). One mechanism is the release of molecules from the EV that interact with surface molecules of the target cells to activate intracellular signaling. For example, researchers found that tumor-cell-released EVs carry programmed death-ligand 1 (PD-L1) on their surface, which interacts with programmed death-1 (PD-1) receptor on T cells to elicit an immune checkpoint [47]. Alternatively, EVs can affect the targeted cells by the internalization and transfer of their cargos. In addition, there are different opinions on the mechanism by which internalization of exosome occurs, such as by membrane fusion, receptor-dependent endocytosis, micropinocytosis, or phagocytosis [48,49]. The latter two mechanisms (i.e., micropinocytosis and phagocytosis) might contribute to the clearance of EVs. Direct evidence of EVs fusing and being endocytosed into recipient cells has been obtained by using lipophilic-dye-labeled EVs, resulting in an increase of the fluorescence of recipient cells. Such real-time imaging technology provides valuable lessons for the study of EV internalization [50].

## 3. EV as Therapeutic Tools for Diabetes and Diabetic Complications

Diabetes and its complications are a significant cause of increase morbidity and mortality worldwide, with an estimated 4.2 million deaths cause by diabetes in 2019 [51]. EVs are a strong candidate as a cell free therapy. The following sections discuss recent updates on the application of EVs in the treatment of diabetes and associated complications.

### 3.1. EV for Type 1 Diabetes

T1D is a chronic disease characterized by insulin deficiency due to autoimmune destruction of the insulin-producing pancreatic islet β-cells, leading to hyperglycemia. The age of symptomatic onset is usually during childhood. Although the mechanism of T1D in still not completely understood, the pathogenesis of the disease is thought to be mediated by abnormalities in multiple immune cells, including T cells, B cells, regulatory T cells (Treg), monocytes and macrophages (Mo/Mϕ), and dendritic cells (DCs) [8]. While multiple daily injections of exogenous insulin allow T1D subjects to manage their blood sugar levels, this does not address the fundamental immune dysfunctions and thus does not represent a cure for T1D. Given their similar function to their parent cells, EVs as therapeutic agents have attracted increased research attention. Kuroda et al. [52] used mesenchymal stem cell (MSC)-derived EVs to treat an autoimmune diabetic mouse model (adoptive transfer T1D mouse model) and found a delay in the onset of T1D by inhibiting T-cell proliferation and suppressing the activation of antigen-presenting cells (APCs). In addition, Nojehdehi et al. [53] claimed that intraperitoneal injection of bone-marrow-derived MSC (BMMSC)-derived exosomes could ameliorate the inflammatory reaction in a streptozotocin (STZ)-induced T1D murine model by increasing regulatory T-cell population. Subsequently, Favaro et al. [54] observed that MSC-derived EVs induced the transformation of T1D patients’ monocytes into immature IL-10-secreting DCs in vitro, potentially contributing to the inhibition of the inflammatory T-cell responses to islet antigens in an STZ-induced T1D rat model. Moreover, these therapeutic EVs not only attenuate inflammation in T1D but also play an important role in β-cell regeneration. Mahdipour et al. [55] claimed that menstrual-blood-derived-MSC (MenSC)-derived exosomes enhanced the β-cell mass and insulin production through the pancreatic and duodenal homeobox 1 (PDX-1) pathway in an STZ-induced T1D mouse model. Tsykita et al. [56] stated that the bone-marrow-cell-derived EVs contain miRNAs (miR-106b-5p and miR-222-3p) that could improve β-cell proliferation by downregulating the CIP/calcium and integrin binding 1 (KIP) pathway. In addition, the EVs displayed a significant role in improving the islet transplantation outcome. Wen et al. [57] applied human (h)BMMSC and peripheral blood mononuclear cell (PBMC) cocultured exosomes, which were more effective than hBMMSC-derived exosomes, to improve islet allograft survival in humanized NOD scid IL-2Rγ ^null^ (NSG) mice. Recently, Sun et al. [58] reported that EVs derived from mouse pancreatic β-cell line MIN-6 could increase insulin content of pancreatic islets and preserve the islet architecture in STZ-induce diabetic mice. Thus, EVs display translational potential to treat T1D and further investigation is required before their clinical application.

### 3.2. EVs for Type 2 Diabetes

Type 2 diabetes (T2D) is the most prevalent form of diabetes and is characterized by two interrelated metabolic defects: insulin resistance combined with pancreatic islet β-cell dysfunctions [59]. T2D results from an interaction between genetic, environmental, emotional, and behavioral risk factors. Previously, hypoglycemic agent administrations and insulin injections were the primary treatments for T2D [60]. Interestingly, recent studies show some EVs might have therapeutic potential to treat T2D. Sun et al. [61] found that intravenous injection of EVs isolated from human umbilical cord MSCs partially reverted insulin resistance via indirectly accelerating glucose metabolism and ameliorating β-cell destruction in STZ-induced diabetic rats with a high fat diet. Mechanistic analysis showed that EV treatment (1) restored phosphorylation of insulin receptor substrate 1 (IRS-1) and protein kinase B in T2D rats, (2) promoted the expression of translocation of glucose transporter 4 (GLUT4) in muscle, and (3) maintained glucose homeostasis via increased storage of glycogen in the liver. In addition, they found that MSC-derived EVs relieved β-cell destruction to restore their insulin secretion in T2D rats. In another study, Zhao et al. [62] treated obese (induced by a high-fat diet) mice with EVs obtained from adipose-derived stem cells (ADSCs) and found that these EVs could polarize macrophages into anti-inflammatory type 2 macrophages (M2) phenotypes by an activation of signal transducer and activator of transcription 3 (STAT3) pathway, which consecutively upregulated arginase 1(ARG-1) expression in macrophage, thereby improving both the metabolic balance and insulin resistance in mice.

Moreover, there is a novel source of stem cells called cord-blood-derived multipotent stem cells (CB-SCs) that are phenotypically different from other types of stem cells, such as hematopoietic stem cells (HSCs), mesenchymal stem cells (MSCs), endothelial progenitor cells (EPCs), and monocyte-derived stem cells [63]. In vitro and animal studies have demonstrated that CB-SCs have a strong therapeutic potential for diabetes [64]. Based on their unique properties of immune modulation and the ability to tightly adhere to the surface of petri dishes, our research group developed a new technology, termed Stem Cell Educator (SCE) therapy, in clinical trials for the treatment of T1D [65,66], T2D [67], and autoimmune-caused alopecia areata [68]. During SCE therapy, the PBMCs are collected, circulated through a cell separator, and cocultured with adherent CB-SCs in vitro. These “educated” cells are then returned to the patient’s circulation via a closed-loop system [68]. Clinical trials have already demonstrated the clinical safety and efficacy of SCE therapy to treat diabetes [65,66,67,69]. Further study of CB-SCs showed that they release exosomes with an immune modulation function similar to that of the original cells, which induce monocytes to differentiate into anti-inflammation type 2 (M2) macrophages [70]. CB-SC-derived exosomes demonstrated not only the mechanism of SCE therapy but also provided attractive materials to treat diabetes.

In addition to these EVs with therapeutic capabilities for diabetes, there are some candidates that could be applied in the future. Human Treg cells are essential to maintain peripheral tolerance, prevent autoimmunity, and limit chronic inflammation [71]. The well-established protocol for the amplification of Tregs in vitro provided researchers with the possibility of isolating EVs from Treg-cultured conditional medium [72,73]. Sistiana et al. [74] demonstrated that EVs derived from Tregs exerted immune suppression on T-cell proliferation and prolonged kidney allograft survival in a mouse model. Tung et al. [75] found that Treg-derived EVs induced DCs to acquire a tolerogenic phenotype, with increased IL-10 and decreased IL-6 production. Additionally, myeloid-derived suppressor cells (MDSCs) are a heterogeneous population of cells that expand during cancer, inflammatory diseases and autoimmune disorders. They have a remarkable ability to suppress immune response [76]. MDSC-derived EVs exhibited an immune-modulation function by inhibiting T-cell proliferation and promoting Treg expansion, which prevented progression and was sufficient for partial hair regrowth in an alopecia areata (AA) mouse model [13]. These findings regarding the immune-modulation function of EVs suggest their therapeutic potential in diabetes.

### 3.3. EV for Diabetic Complications

Diabetic complications are mainly caused by high-glucose-induced cellular and molecular impairments and dysfunctions of neural and cardiovascular systems. Recently, EVs have been recognized as powerful therapeutic candidates for the treatment of this disease. In the following sections, we discuss current research findings for their applications in different diabetic complications.

#### 3.3.1. Diabetic Wounds

In patients with diabetes, the hyperglycemic environment leads to wounds that either heal slowly or fail to heal, thereby posing a serious challenge for healthcare in the clinical setting. The exact pathogenesis of delayed wound healing in patients with diabetes is poorly understood. However, human and animal studies show impairments at different phases of the wound-healing process [77,78]. The risk of infection increases with impairments during wound healing; therefore, accelerating wound healing is an urgent need in diabetes. Chen et al. [79] found that a pro-angiogenic protein called deleted in malignant brain tumor 1 (DMBT1) was enriched in EVs from urine-derived stem cells (USCs). In vivo studies showed that DMTB1 could promote angiogenesis and wound healing in diabetic mice.

Furthermore, MSCs derived from different sources are attractive cells to isolate EVs for therapeutic applications in diabetic wounds. Li et al. [80] reported that EVs derived from nuclear factor erythroid 2 like 2 (NRF2)-overexpressed ADSCs could promote cutaneous wound healing via improving vascularization in a rat model of diabetic foot ulcers. Functional assays demonstrated that EVs reduced inflammation cytokines (IL-6, IL-1β, and tumor necrosis factor alpha (TNF-α)) and oxidative-stress-related proteins to protect EPCs in the wound-healing process in a high-glucose environment. Tao et al. [81] reported that EVs from microRNA-126-overexpressing synovium MSCs (SMSCs) promoted migration and tube formation of HMEC-1 cells in vitro. Functionally, the EVs accelerated diabetic wound healing by promoting re-epithelialization, activating angiogenesis, and developing collagen maturity. Similarly, Ding et al. [82] reported that EVs derived from deferoxamine-stimulated human BMMSCs deliver their exosomal miRNA-126 to downregulate phosphatase and tensin homolog (PTEN), which stimulated angiogenesis in vitro and enhanced wound healing in STZ-induced diabetic rats. Furthermore, Li et al. [83] found that an lncRNA called lncRNA 19H in exosomes derived from MSCs prevented apoptosis and inflammation in fibroblasts by impairing miR-152-3p-mediated PTEN inhibition, leading to the stimulation of wound-healing in a rat model of diabetic foot ulcers. A study reported that treatment combining gingival MSC (GMSC)-derived exosomes with chitosan/silk hydrogel led to more efficient wound healing in a diabetic rat skin defect model than chitosan/silk hydrogel treatment alone. Functionally, the GMSC-derived exosomes promoted skin wound healing by improving the re-epithelialization, angiogenesis, and neuronal ingrowth in diabetic rats [84].

Zhang et al. [85] reported that EVs from umbilical-cord-blood-derived endothelial progenitor cells (UCB-EPCs) that promoted angiogenesis of endothelial cells through activating extracellular regulated kinase (ERK)1/2 signaling. This led to enhanced cutaneous wound repair and regeneration in a diabetic rat model. Similarly, a study claimed that EPC-derived EVs accelerated wound healing in diabetic rats by stimulating endothelial cell proliferation and migration via increasing the levels of angiogenesis-related molecules, such as fibroblast growth factor 1 (FGF-1), vascular endothelial growth factor A (VEGF-A), vascular endothelial growth factor receptor 2 (VEGFR-2), and angiopoietin-1 (ANG-1) [86].

In addition to these stem/progenitor-cell-derived EVs, there are other EVs that could be used to treat diabetic wounds. A study reported that macrophage-derived EVs accelerated wound healing by exerting anti-inflammatory effects and improving endothelial cell function in a diabetic rat model [87]. Guo et al. [88] showed that EVs derived from platelet-rich plasma (PRP) promoted the healing process in a diabetic rat model. Molecular mechanism experiments showed that PRP-derived EVs induced the migration and proliferation of fibroblast and endothelial cells to improve re-epithelialization and angiogenesis via activating the Rho-yes associated protein (YAP) signaling pathway in the diabetic rat model. Geiger et al. [89] found that EVs derived from human fibrocytes accelerated wound healing in diabetic mice. These EVs promoted angiogenesis, activated fibroblasts, and enhanced the function of keratinocytes via carried exosomal cargo molecules, including heat-shock protein 90 alpha (HSP90α), STAT3, proangiogenic miRNAs (miR-126, miR-130a, and miR-132), and anti-inflammatory miRNAs (miR-124a and miR-125b). Furthermore, artificial EVs loaded with specific biomolecules provide a new alternative method for the treatment of diabetic wounds. Researchers found that several biomolecules were significantly reduced in diabetes, including the lncRNA-H19 and miRNA-126. A study stated that artificial EV-mimetic nanovesicles (EMNVs) loaded with lncRNA-H19 have a powerful capacity to restore the regeneration-impairing effects of hyperglycemia and could significantly accelerate wound healing in a diabetic rat model [90]. In summary, these studies used different EVs as therapeutic tools, which effectively promoted diabetic wound healing and could be developed as alternative clinical treatments in the future.

#### 3.3.2. Diabetic Stroke

Diabetes induces a variety of vascular pathologies, including increased vascular permeability, which leads to the elevated morbidity associated with ischemic stroke [91]. Moreover, diabetes alters metabolism and increases inflammation, resulting in complicated stroke pathology and aggravated vascular and white matter damage after stroke, making it more challenging to treat the ischemic brains of patients with diabetes [92]. Venkat et al. [93] reported that treatment with brain endothelial cells (EC)-derived EVs could markedly improve the neurological and cognitive function in type 2 diabetic (T2D)-stroke mice. These improvements might include increasing the densities of axon, myelin, and blood vessels, as well as the polarization of anti-inflammation type 2 macrophage (M2) differentiation. Mechanistic studies demonstrated that EC-derived EVs enrich miR-126 and may contribute to the EV-mediated restoration of neuronal function and axonal outgrowth. The most common diabetic complications in T2D patients are stroke and cardiovascular disease, significantly increasing patients’ mortality risk [94]. Venkat et al. [95] reported that treatment with EVs isolated from human-cord-blood-derived CD133^+^ stem cells could attenuate post-stroke cardiac dysfunction in T2D-stroke mice through decreasing the myocardial cross-sectional area and interstitial fibrosis, downregulation of the level of transforming growth factor beta (TGF-β) and the number of type 1 macrophages (M1), and upregulation of miR-126 expression in the heart of T2D-stroke mice [95].

#### 3.3.3. Diabetic Retinopathy

Diabetic retinopathy (DR) is a severe diabetes mellitus complication and a major cause of vision loss in middle-aged and elderly people. Hyperglycemia is considered to play an important role in the development and progression of DR. DR exhibits the microvascular defects, neuroretinal dysfunctions, and the degeneration of the retina [96,97]. Recently, EVs derived from MSCs showed therapeutic potential to treat diabetic retinopathy. Safwat et al. [98] reported that intraocular or subconjunctival (but not intravenous) injections of EVs derived from adipose-tissue-derived MSCs could protect retinal tissue structures from degeneration in a STZ-induced rabbit model of diabetic retinopathy. Similarly, a study found that intravitreal injection MSC-derived EVs effectively reduced the levels of inflammatory markers IL-1β, IL-18, and caspase-1 in the vitreous humor of STZ-induced diabetic rats. Functionally, MSC-derived EVs containing miRNA-126 play an essential role in reversing the actions of inflammation through the inhibition of high mobility group box 1 (HMGB1). Furthermore, the author found EVs derived from miRNA-126-overexpressed MSCs were more effective in reducing inflammation in diabetic retinopathy [99].

#### 3.3.4. Diabetic Cardiomyopathy

Diabetic cardiomyopathy is defined as diabetes-associated changes in the structure and function of the myocardium which affect around 12% of diabetic patients and result in heart failure and death. Wang et al. [100] reported that HSP20 overexpressing cardiomyocyte-derived EVs containing enriched levels of HSP20 could protect endothelial cells and cardiomyocytes against hyperglycemia-induced stress in vitro. An in vivo functional study found that these HSP20-enriched EVs could ameliorate hyperglycemia-induced cardiac adverse remodeling by significantly improving the left ventricular internal diameter at end-diastole (LVIDd), the left ventricular ejection fraction (LVEF%), and the density of myocardial blood vessels in STZ-treated diabetic mice. A study reported that MSC-derived EVs restored the increased level of left ventricular collagen (LVC) and reduced the expression of fatty acid transporters (FATPs) and fatty acid beta oxidase (FA-β-oxidase) in STZ-induced diabetic rats. Mechanistic analysis showed that MSC-derived EVs inhibited the TGF-β1/SMAD family member 2 (SMAD2) signaling pathway, which played an important role in the EV-associated improvement in diabetes-induced myocardial injury and fibrosis [101].

#### 3.3.5. Diabetic Neuropathy

Diabetic peripheral neuropathy (DPN) is one of the most prevalent chronic complications of diabetes mellitus; it starts with sensory loss in distal nerves [102]. Recently, EVs were identified that could be used to improve neuropathy dysfunction in diabetes. Schwann cells (SCs), the most abundant cells in the peripheral nervous system, interact with axons and blood vessels to regulate peripheral nerve function [103]. An early study demonstrated that SC-derived exosomes markedly increased axonal regeneration in vitro and promoted regeneration after sciatic nerve injury in vivo [104]. Furthermore, Wang et al. [105] reported that EVs derived from SCs (Sc-Exos) remarkably ameliorated DPN by improving sciatic nerve conduction velocity and increasing thermal and mechanical sensitivity in a diabetic mouse model. Functionally, molecular analysis of sciatic nerve tissues showed that SC-Exo treatment reversed diabetes-reduced miR-21, miR-27a, and miR-146a levels, as well as diabetes-increased Semaphorin 6A (SEMA6A), PTEN, and nuclear factor-κB (NF-κB) levels. In addition to Sc-Exos, EVs from stem cells also showed a therapeutic function in DPN. Fan et al. [106] found that treatment of DPN by using MSC-derived EVs corrects neurovascular dysfunction and promotes the functional recovery in diabetic mice, leading to an increased number of intraepidermal nerve fibers, myelin thickness, and axonal diameters of sciatic nerves. Western blotting analysis further revealed that MSC-derived EV treatment reduced the inflammatory response by decreasing the level of M1 and increasing the level of M2 macrophages, respectively.

#### 3.3.6. Diabetic Cognitive Dysfunction

There is increasing evidence linking diabetes to cognitive decline, resulting in dementia among both T1D and T2D. T2D is associated with an approximately 1.5-to 2.5-fold increase in the risk of dementia and has been associated with impaired memory, executive function, attention, processing, and motor speed [107]. Recently, some EVs were identified that could be used to improve cognitive dysfunction in diabetes. Nakano et al. [108] found that intracerebroventricular injection of EVs derived from BMMSCs could improve the diabetes-induced cognitive impairment in a STZ-induced mouse model. Histological analysis showed that, while these EVs did not increase the numbers of neurons, they inhibited oxidative stress and increased the synaptic density in the CA1 region of the hippocampus. Similarly, Zhao et al. [109] showed that intracranial injection of EVs from BMMSCs ameliorated diabetes-induced cognitive disorder, in which the EV-treated group showed a shorter escaping delay in a water-maze experiment in STZ-induced diabetic mice. In addition, a study reported that EVs derived from miRNA-146a-loaded brain endothelial cells injected into the brain ventricles of T2D db/db mice could partially restore short-term memory function and downregulated the expression of prion protein (PrPc), which accumulates in the brain cells of diabetic model mice [110]. These results indicated that EVs might be a promising therapeutic tool to treat diabetes-induced cognitive impairment.

#### 3.3.7. Diabetic Erectile Dysfunction

Erectile dysfunction (ED) is a common and underappreciated complication of diabetes. Studies have shown an increased incidence of ED in patients with diabetes. Moreover, ED appears to symptom about 10 years earlier and is more resistant to treatment in patients with diabetes than in the non-diabetic population [111,112]. A recent study reported EVs derived from stem cells could be applied to ameliorate ED in an animal model. Chen et al. [113] reported that intracavernous injection of EVs from ADSCs could promote the recovery of erectile function by inhibiting apoptosis of corpus cavernosum endothelial and smooth cells in a rat model of diabetic ED. Similarly, a study found that ADSC-derived EVs could rescue the ED in STZ-induced diabetic rats. Bioinformatic analysis found that these EVs contain antifibrotic miRNAs (miR-let7b and miR-let7c) and proangiogenic miRNAs (miR-126, miR-130a, and miR-132), which could downregulate the level of fibrosis and improve angiogenesis in the cavernosum [114]. Interestingly, Ouyang et al. [115] demonstrated that EVs from urine-derived stem cells (USCs) significantly improved erectile function in an STZ-induced diabetic ED rat model. Mechanistic analysis showed these EVs were enriched for a distinct class of angiogenesis-related miRNAs, including miR-21-5p, the let-7 family, the miR-10 family, the miR-30 family, and miR-148a-3p. Additionally, Kwon et al. [116] reported that embryonic-stem-cell-derived extracellular vesicle mimetics (ESC-NVs), which were developed by cells extruded through serial filters with diminishing pore size (10, 5, and 1 μm), could fully restore erectile function in STZ-induced diabetic mice. Histological analysis showed that the ESC-NVs induced neural regeneration in the corpus cavernosum in diabetic conditions and restored cavernous endothelial, smooth muscle cell, and pericyte content. In summary, EVs from stem cells have a positive role in improving diabetic ED and may be applicable to clinic in the future.

#### 3.3.8. Diabetic Nephropathy

Diabetic nephropathy (DN) is one of the most severe complications of diabetes and is the leading cause of end-stage renal disease (ESRD), which is the last stage of chronic kidney disease (CKD) [117]. Currently, hemodialysis and transplantation are used to treat ESRD. Both approaches have limitations, however, including high cost and unpredictable organ availability [118]. Recently, MSC-derived EVs have been recognized as a promising therapy for DN. One early study found that MSCs improved DN through the paracrine effect of renal trophic factors, including EVs in both STZ- and high-fat diet (HFD)-induced diabetic mice. These EVs exerted an antiapoptotic effect and protected tight junction structures in tubular epithelial cells [119]. Similarly, Grange et al. [120] found that the administration of EVs derived from BMMSCs and human liver stem-like cells (HLSCs) significantly improved renal function in diabetic mice. Histological analysis found the renal fibrosis that develops during DN progression was significantly inhibited and reverted in the EV-treated group. Mechanistic analysis showed that these EVs contain certain specific miRNAs that downregulated pro-fibrotic gene expression to inhibit renal fibrosis in DN. Additionally, Ebrahim et al. [121] claimed that EVs from BMMSCs markedly restored renal function in DN by inducing autophagy through the mechanistic target or rapamycin (mTOR) signaling pathway in diabetic rats. Similarly, Jin et al. [122] found that EVs from ADSCs markedly ameliorated DN symptoms via exosomal miR-486, which led to the inhibition of the SMAD1/mTOR signaling pathway in podocytes. In addition, EVs from USCs also play an important role in the treatment of DN. Jiang et al. [123] found that human USC-derived EVs improved renal function by inhibiting podocyte apoptosis and promoting vascular regeneration in a type 1 diabetic rat model. Furthermore, a study found that human USC-derived EVs could protect podocytes in diabetic rats. Mechanistic analysis demonstrated that exosomal miR-16-5p plays an important role in suppressing VEGFA and podocyte apoptosis, thus restoring renal function in DN. Consequently, the application of EVs from miR-15-5p-overexpressing human USCs more effectively restored podocyte function in diabetic rats, which provided fresh insights for novel treatments of DN [124]. These results indicated that EVs might be a promising therapeutic tool to treat DN.

## 4. Ongoing and Completed EV-Based Clinical Trials

To date, practical applications of EVs have been translated into clinical trials for the diagnosis and treatment of a variety of diseases, including diabetes, cancers, infections, and inflammation- or autoimmune-associated diseases [17,18,19,125]. Currently, there are eighteen diabetes-related clinical trials (Table 3) through searching the database of http://clinicaltrials.gov and using keywords such as “diabetes”, “exosome”, “extracellular vesicle”, or “microvesicle”. Due to carrying the mRNAs, miRNAs, lipids, and proteins of their parental cells in EVs, most of these clinical trials are utilizing EVs as biomarkers for clinical diagnosis and monitoring the disease progresses post-treatments (Table 3, clinical trials #1–17). Nassar et al. [126] applied an intra-arterial injection of mesenchymal stem cells (MSC)-derived EVs to treat 20 patients of chronic kidney disease (CKD) at stages III and IV; 10 of the patients were diabetic. The study demonstrated that treatment with EVs significantly improved the estimated glomerular filtration rate (eGFR) and the overall kidney function in grade III–IV CKD patients [126]. Additionally, the same group proposed a clinical trial (NCT02138331, Table 3, #18) to treat type 1 diabetes by using the mesenchymal stem cells (MSC)-derived EVs. There are ongoing clinical trials by using EVs to treat other inflammation- or immune-dysfunction-associated diseases (Table 3, #19–22). MSC-derived EVs may potentially display the similar therapeutic potentials due to containing MSC-released secretome such as cytokines, chemokines, and anti-inflammatory factors [18]. Further optimization with clinical grade EVs is needed to advance the EV-based cell-free therapy.

## 5. Future Considerations

Although EVs display tremendous therapeutic promise, the translation of EVs for clinical usage remains unanswered in several issues.

### 5.1. Large-Scale Production of EV

The translation of EVs into clinical therapy requires large-scale production of clinical-grade EVs. To produce large amounts of EVs, two issues need to be resolved: the production of large quantities of cells and the retention of the cell or EV phenotype [127]. To scale up the cell culture, researchers have used T-flasks as scalable cell culture surfaces or stimulated cells with various stimuli. These techniques, however, might change the composition and function of EVs [31,128,129]. Furthermore, researchers applied 3D culture technology to maximize the surface area for cell culture, such as microcarriers in stirred bioreactors or bioreactors with hollow fiber bioreactor system [130,131]. However, these technologies are limited because changes in the environmental parameters in the reactors will change the phenotype of the cells and their derived EVs. Many factors can influence the quality and quantity of cell culture supernatant-derived EVs: cellular density, early or later passage of cells, oxygen concentration, cytokines or heparin, and medium formulation [125,127]. For example, scientists found that EVs derived from early passage of BMMSC exhibited more efficient neuroprotective potential than later passage derived EVs [132]. Addressing this problem, researchers suggested applying EVs derived from three to five passages of cells for clinical use, as they have similar functions and capabilities as MSC [133,134]. Additionally, medium formulation is a major barrier for EVs clinical translational application. Fetal bovine serum (FBS), for example, with RNA-containing EVs, affects the behavior of cultured cells. Serum-free culture media can change the composition of EVs proteins [135,136]. To elucidate these issues, researchers apply platelet lysate with EV-depleted medium to culture hBMMSCs, retaining the phenotype and differentiation potential of stem cell and ensuring the EV’s RNA profile is unchanged [137]. This protocol provides an alternative for large-scale GMP-based EVs production.

### 5.2. Clinical-Grade EV Isolation and Storage

Currently, there is no state-of-the-art technology to produce EVs in large clinical-scales for therapeutic applications. As mentioned in the EV isolation section, although the ultracentrifugation method is the “gold standard” for exosome isolation, its low yield limits its utility for clinical grade EV isolation. The other methods are also unsuitable for clinical-grade EV isolation because of chemical reagent/antibody contamination or low purity. Recently, researchers have utilized the ultrafiltration to concentrate conditioned medium, followed by size exclusion chromatography (SEC) to obtain EVs [138]. This isolation method offers a higher yield and better preservation of the EVs’ biophysical properties and has attracted increased research attention regarding its clinical application [139].

There is no standard protocol for the storage of isolated clinical-grade EVs for future use. Cryopreservation with cryoprotectants, such as glycerol and dimethyl sulfoxide (DMSO), are not an ideal method for EVs’ storage. One study found that 5% glycerol and 1% DMSO partially or fully lysed the EVs [140]. To date, many groups have used phosphate-buffered saline (PBS) as a storage buffer to conserve EVs’ functional and physical properties. However, the limited amount of calcium included in EVs will lead to the formation of nanosized calcium phosphate particles in PBS, which could interfere with EV quantification [141]. In terms of the storage temperature for EVs, researchers have reported that EVs are more stable when stored at −80 and −20 °C, compared with storage at 4 °C or higher temperatures [140,142]. Alternatively, some researchers applied lyophilization of EVs to extend their shelf life and decrease storage demand and cost. In this method, the best storage temperature stated for lyophilized EVs was 4 °C [143].

### 5.3. Targeting EVs to Cells

The specificity of EVs toward their target cells was demonstrated by Denzer et al. [144]. MHC class II expressed by the EVs adhere to the follicular dendritic cells (FDC) and their released exosomes and could stimulate the proliferation of specific T lymphocytes in vitro, though they are not expressed by FDCs themselves [144]. Likewise, EVs from platelets transfer tissue factor to monocytes and endothelial cells, but not to neutrophils [145]. While the basis of EV targeting is still unclear, some molecular- or cell-dependent targets have emerged. Variations in the presence of recipient cells, EV surface molecules, and the recipient cell’s physiological state mediate the specificity of EV internalization to recipient cells. For example, EVs or receipt cell surface tetraspanins and integrins are important for EV endocytosis, as confirmed when inhibition of these proteins on EVs or recipient cells resulted in reduced EV internalization [146,147].

For EVs to correct a metabolic disorder or promote tissue regeneration in diabetes and diabetic complications, they must be directed and monitored to the correct cells and then internalized. For EVs directed to correct target cells, the method of EVs administration should be addressed during the treatment. For instance, treatment with MSC-EVs by intravenous injection did not restore the dysfunction, but subconjunctival injection fully restored the dysfunction in diabetic retinopathy. The different administration routes of EVs exhibited different therapeutic effects. The majority of these studies were focused on the function changes, without clarifying the exact cellular target by EVs. To trace EVs during ex vivo or in vivo studies, researchers have labeled EVs with lipophilic dyes, including PKH26, PKH67, DIO, and DID, to track the EV targeted cells [148]. Using the fluorescence dye DiO-labeled stem cell CB-SC-derived exosomes, we found that exosomes preferably bound to the CD14^+^ monocytes in human peripheral blood-derived mononuclear cells (PBMC), leading to an upregulation of mitochondrial membrane potential of treated monocytes and differentiation into type 2 macrophages [70]. Another study used PKH-67-labeled EVs and found that their incorporation into skin tissue accelerated wound healing in vivo [50]. These lipophilic dyes offer a powerful tool to study EV targeting and guide EVs selection depending on the specific cell target.

### 5.4. EV Modification

While it is clear that EVs are therapeutically efficacious in several disease models, in order to ensure the delivery of EVs to their sites with therapeutic actions, while minimizing an accumulation at off-target sites, there is strong interest in enhancing EV therapeutic properties, which might allow for lower doses or less frequent administrations. Several methods to increase the therapeutic function of EVs have been applied, such as engineering, priming, loading, and artificial EVs [149]. A study linked EVs to a hydrogel via a photo-cleavable linker, which, upon laser stimulation, triggers the controlled release of exosomes to promote wound closure in a murine model [144]. In addition, a study primed EVs by exposing MSC to the inflammatory cytokines IFN-γ, after which the released EVs were more effective in the context of an acute lung injury model [150]. Furthermore, miRNA-181a was overexpressed to MSC, followed by the collection of the miRNA-181a-abundant exosomes, which showed increased efficacy in treating ischemia-reperfusion injury [151].

To obtain clinical grade EVs on a large scale, researchers propose to design and manufacture fully synthetic EV-mimetic particles by using bionanotechnology. For example, scientists observed that the levels of APO2 ligand (APO2L) (also known as TNF-related apoptosis-inducing ligand (TRAIL)) were drastically decreased in synovial fluid from patients with rheumatoid arthritis. Next, they conjugated APO2L with artificial lipid vesicles resembling exosomes, which downregulated the T-cell activation in an antigen-induced arthritis animal model [152]. Finally, researchers have extruded living embryonic stem cells through a micro-filter, to generate nanovesicles, which show therapeutic potential for wound healing [153].

MSC could not only give rise to multiple cell lineages (e.g., osteoblasts and adipocytes), but could also improve the tissue regeneration, angiogenesis, and anti-inflammation through releasing cytokines, chemokines, growth factors, and EVs [154]. Among these MSC secretomes, MSC-derived EVs have been recognized as a powerful tool that could potentially replace MSC as a cell-free therapy [18]. To advance the therapeutic potential of EVs in drug delivery and regenerative medicine, parental MSC may be genetically modified to produce the growth factor-enriched EVs for the treatment. To this respect, Li et al. [80] reported that exosomes isolated from adipose-derived stem cells (ADSC) could promote the proliferation and angiogenesis of endothelial progenitor cells (EPC). Notably, treatment with exosomes from the genetically-modified ADSC could markedly improve wound healing, the levels of growth factor expressions, and anti-inflammatory effects in diabetic rats after an overexpression of transcription factor nuclear factor-E2-related factor 2 (Nrf2) in ADSC [80], which displayed a protective role against oxidative stress in a diabetic nephropathy animal model [155].

## 6. Conclusions

EVs have demonstrated strong clinical translational potentials for the treatment of diabetes and associated complications (Figure 3). There are some ongoing clinical trials to determine their safety and clinical efficacy. Clinical applications of EVs continue to highlight some practical challenges of using clinical-grade EVs. Further investigations are needed to clarify the composition of EV’s biomolecules (protein, RNAs) from different cell- or tissue-derived EVs for their specific therapeutics, potentially leading to the development of bio-engineered EVs. The mechanistic studies underlying the interaction of EVs with their target cells will facilitate the clinical applications of EVs for the treatment of diabetes and other diseases.

## Figures and Tables

**Figure 1 ijms-21-05163-f001:**
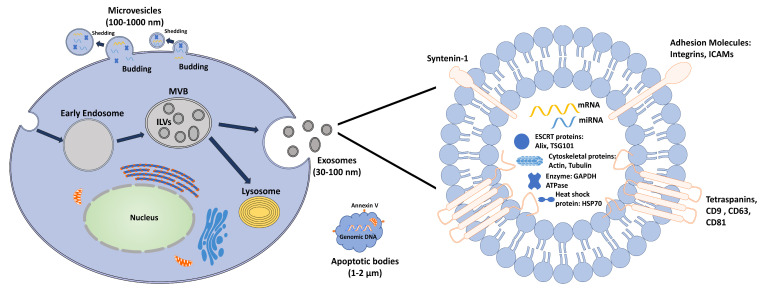
Scheme the biogenesis of EVs. Multivesicular bodies (MVB) are formed during endosomal maturation, and exosomes are released upon fusion of the MVBs with the plasma membrane. Differently, microvesicles are formed directly through cell membrane budding and fission. The apoptotic bodies are derived from the apoptotic cells.

**Figure 2 ijms-21-05163-f002:**
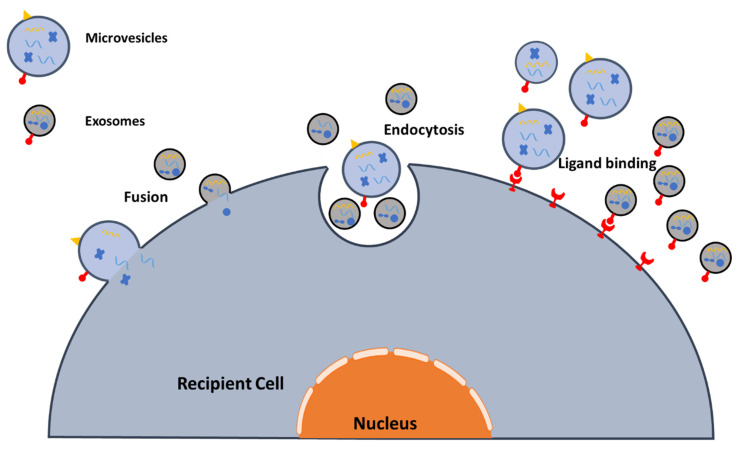
Uptake of EVs. EVs are taken up by the targeted recipient cells via the fusion of the vesicle membrane with the cellular membrane or by endocytosis the receptor and its ligand on EVs.

**Figure 3 ijms-21-05163-f003:**
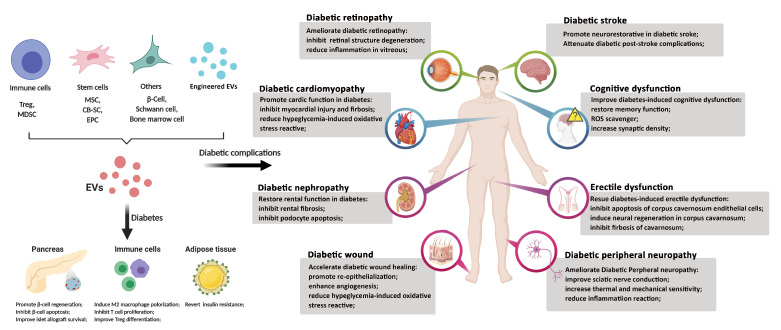
Therapeutic potential of EVs for the treatment of diabetes and complications. This figure was generated with BioRender.com.

**Table 1 ijms-21-05163-t001:** Classification of extracellular vesicles (EVs).

Features of Extracellular Vesicle Subtypes				
	Exosomes	Microvesicles	Apoptotic Bodies	References
Size	30–150 nm	100–1000 nm	1–2 µm	[16,24]
Density	1.12–1.19	1.12–1.21	1.16–1.28	[25,26,27]
Formation	Fusion of multivesicular bodies with the plasma membrane	Outward blebbing of the plasma membrane	Plasma membrane budding of apoptotic cells	[24,28]
Pathways	ESCRT-dependent Tetraspanin-dependent Stimuli-dependent	Ca^2+^-dependent Stimuli- and cell-dependent	Apoptosis-related	[16,24,28]
Content	Protein, mRNA, miRNA, lncRNA, lipid, dsDNA	Protein, mRNA, miRNA, DNA, lipid	Cell organelles, proteins, nuclear fraction, DNA, coding or noncoding RNA, lipid	[16,29,30]
Commonly used markers	CD9, CD63, CD81, Alix, Flotillin-1, ESCRT-3, TSG101	CD40 ligand, selectin, flotllin-2, annexin 1	Annexin V, DNA, histones, phosphatidylserine	[16,25]

**Table 2 ijms-21-05163-t002:** Isolation methods for extracellular vesicles (EVs).

Methods	Principle	Advantage	Disadvantage	Yield	Purity	References
Ultracentrifugation	Size separation; large EVs collect earlier and at lower speed	Cost-effective; no chemical additives	Time consuming; sample aggregation	Low	High	[35,36,39]
Density gradient	Separation by density in sucrose gradient	Cost-effective; no chemical additives	Time-consuming	Low	High	[35,36,39]
Ultrafiltration	Size separation by filtration	Cost-effective; flexible volume; no chemical additives	Time-consuming	high	Low	[36,39]
Polymer-based precipitation	Polymers exclude EVs by altering solubility	Flexible volume; time-saving; no high-cost equipment needed	Polymer and protein contamination; expensive for large sample size	High	Low	[36,39]
Immunoprecipitation	Immobilized antibodies against EVs-specific markers	Time-saving	Expensive; very selective; antibody/protein contamination	Low	High	[36,39]
Size exclusion chromatography	Column-based size separation	Time-saving; no chemical additives	Protein contamination; sample volume limited	High	High	[35,39]
Microfluidic	Microfluidic devices	Flexible volume	Expensive	High	high	[36,40]

**Table 3 ijms-21-05163-t003:** EV-based clinical trials in diabetes and diabetic complications.

No.	Sponsor	Registration No.	Title of Trial	Disease	Status	EV Source
**EV-associated clinical trials in diabetes-associated diagnosis:**						
1	Basque Country University	NCT03027726	Prevention of Diabetes in Overweight/Obese Preadolescent Children	Type 2 Diabetes	Complete	Blood
2	University of Campania	NCT00815399	Pioglitazone Versus Metformin in Type 2 Diabetes	Type 2 Diabetes	Complete	Blood
3	Tan Tock Seng Hospital	NCT01741181	Vitamin D Supplementation in Patients with Diabetes Mellitus Type 2	Type 2 Diabetes	Complete	Blood
4	University of Hull	NCT03102801	A Study to Identify Biomarkers of Hypoglycaemia in Patients with Type 2 Diabetes	Type 2 Diabetes	Complete	Blood
5	Ruhr University of Bochum	NCT02800668	Metabolic Effects of Duodenal Jejunal Bypass Liner for Type 2 Diabetes Mellitus	Type 2 Diabetes	Complete	Blood
6	University Hospital, Clermont-Ferrand	NCT02359461	Evaluation of the Effect of Pulsatile Cuts Stendo3 on Vascular Function Patients with Diabetes Type 2	Type 2 Diabetes	Complete	Blood
7	Kanazawa University	NCT02649465	SGLT2 Inhibitor Versus Sulfonylurea on Type 2 Diabetes with NAFLD	Type 2 Diabetes	Recruit-ing	Blood
8	Shanghai General Hospital	NCT03264976	Role of the Serum Exosomal miRNA in Diabetic Retinopathy	Type 2 Diabetes	Not yet	Blood
9	George Washington University	NCT03660683	Effect of Saxagliptin and Dapagliflozin on Endothelial Progenitor Cell in Patients with Type 2 Diabetes Mellitus	Type 2 Diabetes	Recruit-ing	Urine
10	Centre Hospitalier Universitaire de Nice	NCT02768935	Macrophage Phenotype in Type 2 Diabetics After Myocardial Infarction and the Potential Role of miRNAs Secreted	Type 2 Diabetes	Recruit-ing	Blood
11	Assistance Publique—Hôpitaux de Paris	NCT03634098	Identification and Validation of Noninvasive Biomarkers of the Diagnosis and Severity of NASH in Type 2 Diabetics	Type 2 Diabetes	Recruit-ing	Blood
12	McGill University Health Centre	NCT03106246	Circulating Extracellular Vesicles Released by Human Islets of Langerhans	Type 1 Diabetes Type 2 Diabetes	Recruit-ing	Blood
13	Centre d’Etudes et de Recherche pour l’Intensification du Traitement du Diabète	NCT00934336	Importance in Type 1 Diabetes Patients of an Optimized Control of Post-Prandial Glycaemia on Oxidant Stress Prevention	Type 1 Diabetes	Complete	Blood
14	Karolinska Institutet	NCT01497912	Treatment Effects of Atorvastatin on Hemostasis and Skin Microcirculation in Patients with Type 1 Diabetes	Type 1 Diabetes	Complete	Blood
15	Translational Research Institute for Metabolism and Diabetes	NCT03971955	Characterization of Adult Onset Autoimmune Diabetes	Type 1 Diabetes	Recruit-ing	Blood
16	Mayo Clinic	NCT03392441	Insulin Deprivation on Brain Structure and Function in Humans with Type 1 Diabetes	Type 1 Diabetes	Recruit-ing	Blood
17	Translational Research Institute for Metabolism and Diabetes	NCT04164966	Development of Novel Biomarkers for the Early Diagnosis of Type 1 Diabetes	Type 1 Diabetes	Recruit-ing	Blood
**EV-associated clinical trial in diabetic treatment:**						
18	General Committee of Teaching Hospitals and Institutes, Egypt	NCT02138331	Effect of Microvesicles and Exosomes Therapy on β-cell Mass in Type I Diabetes Mellitus	Type 1 Diabetes	N/A	MSC
**EV-associated clinical trials to treat other inflammation or autoimmune diseases:**						
19	Zhongshan Ophthalmic Center, Sun Yat-sen University	NCT04213248	Effect of UMSCs Derived Exosomes on Dry Eye in Patients with cGVHD	Dry eye syndrome in cGVHD patients	Recruit-ing	MSC from umbil-ical cord
20	Aegle Therapeutics	NCT04173650	MSC EVs in Dystrophic Epidermolysis Bullosa	Dystrophic Epidermoly-sis Bullosa	Recruit-ing	MSC
21	Beni-Suef University	NCT04270006	Evaluation of Adipose Derived Stem Cells Exo. in Treatment of Periodontitis (exosomes)	Periodontitis	Recruit-ing	MSC
22	University Medical Centre Ljubljana	NCT04281901	Efficacy of Platelet- and Extracellular Vesicle-Rich Plasma in Chronic Postsurgical Temporal Bone Inflammations	Chronic Postsurgical Temporal Bone Inflamma-tions	Active, not recruit-ing	Plasma

NAFLD, non-alcoholic fatty liver disease; NASH, non-alcoholic steatohepatitis; MSC, mesenchymal stem cell; UMSCs, umbilical mesenchymal stem cell; cGVHD, chronic graft versus host disease.

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
