# Peer review of "Therapeutic Potentials of Extracellular Vesicles for the Treatment of Diabetes and Diabetic Complications"

_ijms, 2020, doi:10.3390/ijms21145163_

Round 1
Reviewer 1 Report
The review submitted by Hu et al., entitled “Therapeutic potentials of extracellular vesicles for the treatment of diabetes and diabetic complications”, is an excellent article that summarizes a novel approach for treating diabetic patients. This review offers an updated and interesting knowledge regarding the use of extracellular vesicles by thoroughly analyzing the benefits of such therapy on diabetes and a wide range of its associated complications.
The Introduction and objectives of the review are well described. The distribution of the data throughout the review, as well as its Discussion, is meaningful and adequately explained. Likewise, its Conclusion summarizes satisfactorily the main concepts of such a review. Besides, the Figures and Tables appropriately reinforce the information. Noteworthy, the paper is supported by meaningful and updated references.
Moreover, the manuscript is readable by offering a high quality of the English language and a good coherence of the arguments.
Author Response
Dear Reviewer,
We appreciate your kind consideration and very nice comments on this review article.
Best regards,
Yong Zhao
Reviewer 2 Report
This is an interesting review regarding the therapeutic potential of extracellular vesicles for the treatment of diabetes and diabetic complications.
I would suggest to complete by referring to eventual trials ongoing on the use of EVs for the treatment of diabetes nad its complications.
I would summarize the Introduction on EVs (paragraphs 2 .1, 2.2 and 2.3) by referring to published reviews to avoid overlapping i.e Eleuteri S I J Mol Sci 2019 20(18): 4597: Fierabracci A Cell Transplat 2015 24: 133-49 and Others.
I suggest also to refer to specific effects of the EVs secretome.
Author Response
Dear Reviewer,
We appreciate your kind consideration and comments that are very helpful for us to improve the quality of this article. We have provided responses to your questions and comments point-by-point. Please see our responses in the Attachment. Many thanks!
Best regards,
Yong Zhao, MD, PhD

Round 2
Reviewer 2 Report
The article can be accepted in the present format